# Effect of Nutritional Supplementation on Oxidative Stress and Hormonal and Lipid Profiles in PCOS-Affected Females

**DOI:** 10.3390/nu13092938

**Published:** 2021-08-25

**Authors:** Pallavi Dubey, Sireesha Reddy, Sarah Boyd, Christina Bracamontes, Sheralyn Sanchez, Munmun Chattopadhyay, Alok Dwivedi

**Affiliations:** 1Department of Obstetrics and Gynecology, Texas Tech University Health Sciences Center El Paso, El Paso, TX 79905, USA; sireesha.reddy@ttuhsc.edu (S.R.); sarboyd@ttuhsc.edu (S.B.); christina.bracamontes@ttuhsc.edu (C.B.); sheralyn.sanchez@ttuhsc.edu (S.S.); 2Graduate School of Biomedical Sciences, Texas Tech University Health Sciences Center at El Paso, El Paso, TX 79905, USA; munmun.chattopadhyay@ttuhsc.edu (M.C.); alok.dwivedi@ttuhsc.edu (A.D.); 3Department of Molecular and Translational Medicine, Texas Tech University Health Sciences Center El Paso, El Paso, TX 79905, USA; 4Division of Biostatistics & Epidemiology, Department of Molecular and Translational Medicine, Paul L. Foster School of Medicine, Texas Tech University Health Sciences Center El Paso, El Paso, TX 79905, USA

**Keywords:** nutrition, supplement, oxidative stress, PCOS, disease markers

## Abstract

Polycystic ovary syndrome (PCOS) affects several reproductive and endocrine features in females and has a poorly understood etiology. Treatment strategies for PCOS are limited and are based primarily on diet and nutrient supplementation. Recent studies have recommended some nutrients such as vitamins, minerals and vitamin-like nutrients for the therapy for PCOS. Therefore, it is claimed that the cause of PCOS could be vitamin or mineral deficiency. This review provides a narrative on the effect of nutritional supplementation on oxidative stress induced in PCOS. Oxidative stress plays a formative role in PCOS pathophysiology. This article reviews oxidative stress, its markers, nutritional supplementation and clinical studies. We also aim to show the effect of nutritional supplementation on genes affecting hormonal and glucose-mediated pathways.

## 1. Introduction

Polycystic ovary syndrome (PCOS) is a heterogeneous disorder leading to adverse health outcomes across all ages [1]. PCOS is the leading cause of anovulatory infertility and may also lead to major cardiovascular events [2,3]. PCOS is marked by hyperandrogenism and insulin resistance (IR) across all ages and ethnicities [4].

Oxidative stress has been recognized in various pathological disorders related to IR [5], obesity [6], PCOS [7], type 2 diabetes and cardiovascular diseases [8]. The development of oxidative stress occurs when there is a disproportion between the generation of free radicals and antioxidants in biological systems. This disproportion results in a negative equilibrium shift [9]. Free radicals interact with micro- and macromolecules in the body by oxidation and deleterious alterations that occur intra- and extracellularly [10]. Oxidative stress ensues, causing DNA and protein damage. Normal reactive oxygen species (ROS) participate in embryonic processes involving fetal and placental development along with normal oocyte maturation and folliculogenesis [11], whereas excessive oxidative stress can result in fetal growth restriction (FGR), miscarriages or fetal death.

In the disease states associated with oxidative stress, hyperglycemia and increased free fatty acids induce the generation of ROS and impair insulin action. Factors in PCOS that increase oxidative stress are obesity, IR and hyperglycemia; however, non-obese PCOS women without IR are also reported to have increased oxidant status, suggesting that other factors may contribute to inducing the production of ROS in these women. Although the role of oxidative stress in PCOS pathogenesis in unclear, studies show that insulin resistance and oxidative stress cause an interplay affecting PCOS pathophysiology. In this article, we aim to explore the relationship between vitamin supplementation and PCOS-related symptoms.

## 2. Oxidative Stress in PCOS

Abnormal oxidation status can be correlated with diabetes, cardiovascular diseases, PCOS and cancer. OS impacts PCOS pathogenesis leading to insulin resistance, androgen excess and chronic inflammation. Primary circulating markers of oxidative stress include homocysteine, malondialdehyde (MDA), asymmetric dimethylarginine (AMDA), superoxide dismutase (SOD), glutathione (GSH), and paraoxonase 1 (PON1) [10]. Women with PCOS have abnormal levels of circulating biomarkers of OS. The imbalance in the total serum antioxidant level in women with PCOS exaggerates the cellular injury and decreases defense mechanisms. Oxidative stress is affected both by genetic and environmental factors. Increased ROS production potentially disrupts mitochondrial DNA (mtDNA) and induces cell apoptosis. Single-point mutations of genes encoding mitochondrial transfer RNA (mt-tRNA) associated with metabolic complications of PCOS, such as diabetes and hypertension, have been found in some studies [12].

## 3. Markers of Oxidative Stress

The pathogenesis of PCOS can be well studied by observing the biomarkers MDA, NO, and the anti-oxidative biomarkers, primarily TAC, SOD, glutathione peroxidase (GPx) and glutathione stimulating hormone (GSH). The role of oxidative stress and antioxidant biomarkers is to estimate the risk of PCOS severity and cardiovascular events.

### 3.1. Malondialdehyde

Lipid peroxidation of polyunsaturated fatty acids generates malondialdehyde [13]. Murri et al. performed a meta-analysis to show increased levels of MDA concentrations in PCOS females compared to controls [10]. Kuscu et al. and Zhang et al. showed that MDA levels were increased in PCOS patients but were independent of obesity [9,14]. Dursun et al. compared PCOS patients with BMI and smoking status-matched controls and found similar serum MDA levels [15].

### 3.2. Nitric Oxide

NO (Nitric Oxide) is biosynthesized by various nitric oxide synthase (NOS) enzymes from L-arginine, oxygen and nicotinamide adenine dinucleotide phosphate and plays a significant role in cellular signaling. Monocytes, macrophages and neutrophils also synthesize small amounts of NO in immune responses. Murri et al. found no statistically significant differences in NO levels in PCOS subjects vs. controls [10]. Ovaries of Wistar rats treated with L-arginine, an NO precursor, had polycystic characteristics which emphasized the role of NO in the pathophysiology of PCOS [16]. Willis et al. compared NO metabolites in PCOS patients and age/BMI matched controls, showing similar nitrite but lower nitrate levels (nitrite/nitrate ratio = endothelium NO concentration) in PCOS patients (Nacul et al. suggested that NO was associated with insulin resistance in PCOS patients [17].

### 3.3. Total Antioxidant Capacity

Total antioxidant capacity (TAC) estimates the antioxidant response against the free radicals produced in a given disease. Murri et al. meta-analysis showed that TAC levels were similar in PCOS subjects and controls [10]. Fenkci et al. demonstrated that TAC levels were significantly lower in PCOS patients compared to BMI and smoking status-matched controls [18], contrary to Verit et al. who reported higher TAC levels in PCOS patients compared with age and BMI-matched controls. Due to the incongruous nature of the studies, further studies should be conducted to clarify the PCOS–antioxidants interconnection.

### 3.4. Reduced Glutathione (GSH)

Glutathione is an antioxidant produced by the cells to withstand oxidative stress. Glutathione exists in a ratio of reduced (GSH) and oxidized forms (GSSG) to maintain homeostasis [19]. GSH plays an important role in the regulation of the disulfide bonds of proteins and the disposing of electrophiles and oxidants.

Murri et al. revealed that mean GSH levels were 50% lower in women with PCOS than in controls [10], similar to the studies conducted by Sabuncu et al. [20] and Dincer et al. citing the role of increased production of ROS in GSH depletion [21].

## 4. Role of Supplementation in PCOS and Associated Comorbidities

Nutritional supplementation plays a major role in PCOS-related oxidative stress and its associated comorbidities. Over the years, several authors have published a variety of nutritional supplements (Table 1) to alleviate oxidative stress and its associated adverse effects on the hormonal and lipid profiles of females (Table 2).

### 4.1. Vitamin D

The molecular mechanism between vitamin D and PCOS is unclear. Irani et al. reported improvement in the biochemical parameters of women with PCOS when supplemented with vitamin D3, suggesting that the inflammatory progress in the pathogenesis of PCOS may be prevented by vitamin D3. In contrast, Jafari-Sfidvajani et al. found no significant differences in androgen profiles after vitamin D supplementation except improvement in menstrual frequency [50]. A study by Krul-Poel et al. [51] confirmed the role of vitamin D in adverse metabolic events of PCOS. Another study concluded that obese PCOS women had significantly decreased 25-dehydroxy vitamin D levels [52]. Further, lower vitamin D levels were linked with insulin resistance owing to the complex pathophysiology of PCOS [53]. A meta-analysis conducted by Akbari et al. on seven randomized controlled trials (RCT) concluded that vitamin D supplementation in women with PCOS resulted in an improvement in high-sensitivity C-reactive protein (hs-CRP), malondialdehyde (MDA) and total antioxidant capacity (TAC) but did not affect nitric oxide (NO) and total glutathione (GSH) levels [49]. Another RCT with vitamin D-K-calcium showed a trend towards a decline in luteinizing hormone, whereas there was no significant effect of vitamin D-K-calcium co-supplementation on prolactin, follicle-stimulating hormone, 17-OH progesterone, inflammatory markers and glutathione levels [38]. In an RCT, the vitamin D and omega-3 fatty acid co-administration for 12 weeks had beneficial effects on mental health parameters, serum total testosterone (TT) levels, hs-CRP, TAC and MDA in women with PCOS [41]. In another meta-analysis study where reports of RCT and 956 subjects were identified, the results demonstrated that supplementation with vitamin D in PCOS patients improved TT, hs-CRP, TAC and MDA levels without affecting other hormonal and oxidative markers, primarily NO and GSH levels. Low-dose vitamin D supplementation (≤1000 IU/day) on a daily basis was more promising for improving hormones and oxidative stress in PCOS patients [54]. Our observation that vitamin D improves the levels of oxidative stress in PCOS is consistent with the meta-analysis studies of Xue [55] and Lagowska [48]. A study by Kyei et al. found that a combination of vitamin D3 and MitoQ10 significantly reduced the oxidative markers SOD and MDA and reduced the hormonal markers estradiol, progesterone, FSH, LH and LH/FSH. The study also observed an improvement in the histomorphological features of ovaries with many healthy follicles and few atretic follicles after co-administration [43].

### 4.2. Flavonoids and Isoflavones

Both flavonoids and isoflavones consist of polyphenols with antioxidant, antidiabetic and anti-inflammatory properties [56]. Oh et al. analyzed six flavonoid classes for treatment of metabolic syndrome in PCOS and found that only flavonol consumption was effective against metabolic syndrome in PCOS [17]. Romualdi et al. showed an improved lipid profile when administering 36 mg per day of soy isoflavone genistein for 12 weeks to women with PCOS. They observed markedly improved total cholesterol and reduced LDL. No change was observed in anthropometric features, the hormonal milieu, menstrual cycles and overall glycoinsulinemic metabolism [57]. Another study observed the outcomes of a soy diet comprising of, i.e., 0.8 g of protein (35% animal proteins, 35% soy protein and 30% vegetable proteins) per kg^−1^ of body weight in which there was found to be a depletion in body mass index, TT, TGA and MDA levels, while the NO and GSH levels were found to be elevated compared to the controls [36]. Jamilian et al. explored the effects of soy isoflavones on metabolic status of PCOS patients and found that 50 mg per day significantly decreased serum insulin and HOMA-IR, free androgen index, serum TGA and MDA and increased plasma total glutathione compared to the placebo group.

### 4.3. Selenium

Selenium (Se) is protective against oxidative stress [58] and is imperative for reproductive tissue formation [59]. High androgen levels, free radicals and insufficient selenium levels have been associated with women with PCOS [33]. Hajizadeh-Sharafabad et al. conducted a systematic review of seven studies and proposed that Se supplementation decreases the BMI and weight of PCOS patients [60]. Additionally, 200 micrograms (μg) per day of Se plus 8 × 10^9^ colony forming units (CFU)/day probiotic supplementation in PCOS women for 12 weeks reduced body weight and cardio-metabolic adverse outcomes [32]. In contrast, Hosseinzadeh et al. indicated that women with PCOS administered with 200 μg/day of Se for 12 weeks did not show weight change. They also indicated that the indiscriminate consumption of selenium may worsen insulin resistance in PCOS subjects [31]. The findings may be inconsistent due to differences in interventional duration, clinical features of subjects, levels of physical activity and diet. Additionally, Se affects insulin concentrations resulting in reduced body weight, reduced BMI, IGF (insulin-like growth factors) and their binding proteins. A study by Coskun et al. revealed that women with PCOS had lower serum levels of Se and increased insulin levels and HOMA-IR, but, when compared with the controls, the levels were not statistically significant [33]. Shabani et al. indicated that 200 μg/day of Se and 8 × 10^9^ CFU/day probiotics co-supplementation for 12 weeks led to a significant reduction in weight, serum insulin levels and the homeostatic model of assessment for insulin resistance and a significant increase in the quantitative insulin sensitivity check index [61]. In addition, selenium and probiotic co-supplementation reduced serum triglycerides, total LDL and total-/HDL-cholesterol ratio compared with the placebo [61]. Similarly, Razavi et al. reported that 200 μg of Se daily for 8 weeks reduced serum dehydroepiandrosterone (DHEA) levels, decreased acne and hirsutism, and caused higher pregnancy incidence when compared to the placebo [30].

### 4.4. Probiotics

Probiotics have a synergic relationship with gut microbiota, thus having a beneficial effect on metabolism [62,63,64]. Studies on probiotic consumption showed improvement in fasting blood glucose levels and antioxidant status in patients with type 2 diabetes [65] and delayed onset of glucose intolerance, hyperglycemia, hyperinsulinemia and dyslipidemia in diabetic rats [66]. Shoaei et al. found reduced fasting blood sugar and serum insulin levels but unaffected CRP in patients administered with multispecies probiotics for 8 weeks [67].

Inflammation and insulin resistance in PCOS is associated with the dysbiosis of gut microbiota (DOGMA) [68], which affects PCOS pathophysiology and insulin receptor function. Probiotic supplements are recommended to overcome dysbiosis [47]. Fecal microbiota transplantation (FMT) and lactobacillus transplantation in rats with PCOS showed that rats in the FMT group had improved estrous cycles, while most of the lactobacillus-treated rats showed decreased androgen biosynthesis [69].

Nasri et al. reported that synbiotic supplementation enhanced serum sex hormone-binding globulin (SHBG) and plasma NO and decreased modified Ferriman–Gallwey (mFG) scores, FAI, serum hs-CRP, serum insulin levels and HOMA-IR, while no significant effect was observed on hormonal status and biomarkers of oxidative stress [35].

### 4.5. Vitamin E, Folate and Omega-3 Fatty Acids

Vitamin E is a free radical scavenger that synchronizes the oxidant/antioxidant ratio. Cicek et al. concluded in their study that vitamin E could improve features of endometrial lining in infertile women because of its anticoagulant and antioxidant effects [70].

Bahmani et al. showed supplementation of folate (5 mg/day) resulted in reduced plasma Hcy, HOMA-B, serum hs-CRP and plasma MDA concentrations and increased plasma TAC and GSH levels compared with the placebo groups [22].

Omega-3 fatty acids and vitamin E co-supplementation downregulated the gene expression of lipoprotein (a), mRNA and oxidized-LDL mRNA and significantly reduced serum TGA, VLDL, LDL- and total-/HDL cholesterol in PCOS subjects. Plasma TAC levels were increased with a marked decrease in malondialdehyde levels compared with the placebo group [23].

A study conducted by Mirmasoumi et al. in which flaxseed oil omega-3 supplementation was administered for 12 weeks significantly decreased insulin values, HOMA-IR and mFG scores and increased the quantitative insulin sensitivity check index. It also showed a decrease in serum TGA, VLDL-cholesterol and high-sensitivity C-reactive protein (hs-CRP) when compared to the placebo group. No significant effects were observed for flaxseed oil omega-3 supplementation on hormonal profiles, lipid profiles and plasma NO levels [24]. Omega-3 fatty acid intake was found to lead to significantly decreased serum insulin levels, HOMA-IR, TT and hirsutism while increasing the quantitative insulin sensitivity check index (QUICKI). It also reduced high sensitivity C-reactive protein and MDA and enhanced plasma total glutathione when compared to the placebo. Changes in other metabolic parameters were not observed [25].

## 5. Inflammation in PCOS

Insulin resistance, oxidative stress and inflammation are frequently tied to PCOS and also described as the risk factors for the development of metabolic syndrome [71]. A number of inflammatory mediators and chemokines also play important roles in the function of the various female reproductive organs [72,73]. Oxidative stress markers are often associated with increased inflammation. An elevated level of CRP is one of the distinct inflammatory markers that is observed in PCOS individuals, and this phenomenon is also abundantly found in patients with other metabolic syndromes [74]. Interleukin (IL)-18 and monocyte chemoattractant protein1 (MCP-1) are increased in patients with PCOS [75,76]. Patients with PCOS have shown elevated levels of white blood count (WBC) in many studies [77]. The recruitment and activation of leukocytes implemented by the involvement of chemokines such as C-C motif ligand 3 (CCL3) can also lead to metabolic syndrome in PCOS individuals [78]. PCOS and inflammation may be related to low-grade chronic infection. Endothelial inflammation along with markers of endothelial function such as endothelin-1 (ET-1), soluble intercellular adhesion molecule-1 (sICAM-1) and soluble vascular cell adhesion molecule 1 (sVCAM-1) have been found to be elevated in women with PCOS [79,80], and women with PCOS and insulin resistance often showed endothelial dysfunction. Women with PCOS exhibited a state of chronic low-grade inflammation, which may help evaluate the changes in biomarkers and their association with cardiac complications.

## 6. Effective vs. Less Effective Supplementation

Significant markers of oxidative stress include: reduced levels of nitric oxide (NO), low total antioxidant capacity (TAC) and higher malondialdehyde (MDA). A meta-analysis by Chan Meng showed that PCOS was associated with serum/plasma nitrite levels. Furthermore, in patients with PCOS, serum or plasma nitrite levels were reduced compared with controls. Based on previous studies, it has been observed that synbiotic supplementation and dietary soy were beneficial in the overall improvement of nitric oxide (NO) levels in patients with PCOS, but other supplementation seemed to have no effect on NO levels. On the contrary, TAC and MDA levels were improved with carnitine supplementation with and without chromium, melatonin, dietary soy, probiotics and vitamin D supplementation. Selenium supplementation should be taken with caution as it may lead to an increased insulin resistance as shown by some studies. Hirsutism, being one of the major consequences of PCOS, was observed and measured in almost all these clinical trials. The supplements which had an overall favorable effect on hirsutism were melatonin supplementation, inositol and selenium supplementation with probiotics. Modified Ferriman¬–Gallaway (mFG) scores, another PCOS indicator observed in these studies, were significantly improved by probiotics and selenium supplementation. A sharp decline in levels of total testosterone was observed with omega-3 fatty acids, carnitine and chromium co-supplementation, melatonin, probiotics, selenium and vitamin D. The studies that failed to show significant positive effects on the improvement of either symptoms or lipid and hormonal assay included supplementation of alpha lipoic acid, curcumin, vitamin E and hydroxysafflor yellow A.

## 7. Effect on Gene Expression

Some studies targeted the overall effect of supplements on targeted genes. Forty infertile PCOS subjects were administered 200 µg selenium for 8 weeks which resulted in significantly increased expression levels of PPAR-γ and GLUT-1 and decreased expression levels of LDLR [81]. Omega-3 EPA in doses ranging 25–200 µg in granulosa cell cultures resulted in higher IGF-1 expression and lower COX2 expression, which is essential for follicular differentiation and oocyte maturation [82]. Curcumin supplementation significantly increased serum activity of GPx but was statistically insignificant for SIRT1 gene expression [46]. Gene expression of interleukin-1 (IL-1) and tumor necrosis factor alpha (TNF-α) was significantly downregulated with melatonin supplementation in PCOS subjects vs. placebo [45]. Vitamin D MitoQ10 supplementation resulted in significantly low expression rates of mRNAs of 3β-HSD, Cyp19a1, Cyp11a1, StAR, Keap1, HO-1 and Nrf2 [43].

## 8. Conclusions

To summarize, oxidative stress and antioxidant biomarkers can effectively modify the risk of PCOS severity and cardiovascular events in women. Simple nutritional supplements may offset these risks by counteracting the effects of oxidative stress on PCOS. Vitamin, minerals, probiotic supplements and other dietary additives can be significantly beneficial in reducing PCOS-related symptoms.

## Figures and Tables

**Table 1 nutrients-13-02938-t001:** Studies performed on supplementation in human and mice models.

Supplement Composition	Year	Country	Animal/Human Studies	Number of Patients/Animals	Age Group	Type of Study
Folate supplementation [22]	2014	Iran	Human	69	18–40	Randomized, double-blind, placebo-controlled
Omega-3 fatty acids and vitamin E [23]	2017	Iran	Human	68	18–40	Randomized double-blind, placebo-controlled
Flaxseed oil omega-3 fatty acids [24]	2017	Iran	Human	60	18–40	Randomized double-blind, placebo-controlled
Fish oil omega-3 fatty acid [25]	2018	Iran	Human	60	18–40	Randomized double-blind, placebo-controlled trial
Vitamin E [26]	2020	China	Human	321		Retrospective cohort clinical trial
Alpha lipoic acid [27]	2010	USA	Human	6	23–34	NR
Carnitine supplementation [28]	2017	Iran	Human	60	18–40	Randomized, double-blind, placebo-controlled trial
Selenium supplementation [29]	2017	Poland	Human	59	14–18	NR
Selenium supplementation [30]	2015	Iran	Human	64	18–40	Randomized double-blind, placebo-controlled
Selenium supplementation [31]	2016	Iran	Human	53	18–42	Randomized, double-blind and placebo-controlled trial
Probiotic and selenium co-supplementation [32]	2018	Iran	Human	60	18–40	Randomized, double-blinded, placebo-controlled clinical trial
Selenium supplementation [33]	2015	Iran	Human	70	18–40	Randomized, double-blind, placebo-controlled trial
Soy isoflavones [34]	2018	Iran	Human	70	18–40	Randomized, double-blind, placebo-controlled trial
Synbiotic supplementation [35]	2018	Iran	Human	60		Randomized, double-blind, placebo-controlled trial
Dietary soy [36]	2018	Iran	Human	60	18–40	Randomized, double-blind, placebo-controlled trial
Probiotic supplementation [37]	2018	Iran	Human	60	18–40	Randomized, double-blind, placebo-controlled trial
Vitamin D-K-calcium co-Supplementation [38]	2016	Iran	Human	60	18–40	Randomized, double-blind, placebo-controlled trial
Vitamin D [39]	2017	Iran	Human	60	20–40	Case-control study
Vitamin D and evening primrose oil [40]	2017	Iran	Human	60	18–40	Randomized double-blind, placebo-controlled trial
Vitamin D and omega-3 co-supplementation [41]	2018	Iran	Human	60	18–40	Randomized double-blind, placebo-controlled trial
Vitamin D and probiotic co-supplementation [42]	2019	Iran	Human	60	18–40	Randomized double-blind, placebo-controlled trial
MitoQ10 and vitamin D3 [43]	2020	Iran	Mouse	48		NR
Vitamin D [44]	2021	Iran	Mouse	40		NR

NR: not reported.

**Table 2 nutrients-13-02938-t002:** Effect of supplementation on clinical and laboratory features in human subjects.

Supplement	Dosage	Hormonal Changes	Changes in Lipid Profile	Changes in Oxidative Stress Markers	Genes Affected
Folate [22]	5 mg/d for 8 weeks			↓ Plasma Hcy, MDA, serum hs-CRP; ↑ plasma TAC and total GSH levels	
Omega-3 fatty acids and vitamin E [25]	Co-supplementation for 12 weeks		↓ Serum TGA, VLDL, LDL- and total-/HDL cholesterol in PCOS subjects	↑ Plasma TAC levels, ↓ malondialdehyde levels	Gene expression of Lp(a) and Ox-LDL
Flaxseed oil omega-3 fatty acids [24]	Supplementation for 12 weeks	↑ mFG scores	Beneficial effects on insulin metabolism, serum triglycerides, VLDL-cholesterol and hs-CRP levels		
Fish oil omega-3 fatty acid [25]	Supplementation for 12 weeks	Beneficial effects on mental health parameters, total testosterone, hirsutism	Effective on insulin markers	↓ Inflammatory markers and oxidative stress	
Vitamin E [26]	100 mg/d	No significant differences of ovulation rate, clinical pregnancy rate and ongoing pregnancy rate		↓ Oxidative stress	
Alpha lipoic acid [27]	600 mg twice daily for 16 weeks			No improvement in serum oxidative stress markers	
Carnitine and chromium co-supplementation [28]	1000 mg/d carnitine plus 200 mg/d chromium as chromium picolinate for 12 weeks	↓ Total testosterone and hirsutism		↓ Malondialdehyde (MDA) levels and higher total antioxidant capacity (TAC)	↑ Gene expression of interleukin-6 (IL-6) and tumor necrosis factor alpha (TNF-a)
Melatonin supplementation [45]	5 mg melatonin supplements	↓ Hirsutism, serum total testosterone		↓ Malondialdehyde (MDA) levels, ↑ plasma total antioxidant capacity (TAC) levels and total glutathione (GSH)	↓ Gene expression of IL-1, (TNF-α)
Curcumin supplementation [46]	1500 mg/d Curcumin for 12 weeks			↑ Serum activity of GPx	↑ SIRT1 gene expression
Selenium supplementation [30]	200 μg selenium per day for 8 weeks	↑ Pregnancy rate, ↓ Alopecia and acne, ↓ Serum (DHEAS), hirsutism (modified Ferriman–Gallwey scores)			
Probiotic and selenium co- supplementation [32]	8 × 10^9^ CFU/d probiotic 200 μg/d	↓Total testosterone, hirsutism		↓ MDA levels, ↑ TAC and GSH levels	
Selenium supplementation [29]	200 mcg/d for 8 weeks		↓ Serum triglycerides and VLDL-C concentrations	↓ Serum insulin levels, homeostasis model of assessment-insulin resistance (HOMA-IR), (HOMA-B); ↑quantitative insulin sensitivity check index (QUICKI)	
Soy isoflavones [34]	50 mg/d soy isoflavones for 12 weeks	↓ Circulating serum levels of insulin and HOMA-IR, ↑quantitative insulin sensitivity check index, ↓ free androgen index and serum triglycerides, ↑ plasma total glutathione,↓malondialdehyde	↓ Free androgen index and serum triglycerides	↓ Circulating serum levels of insulin and HOMA-IR, ↑ Quantitative insulin sensitivity check index, ↑ Plasma total glutathione ↓Malondialdehyde levels	
Synbiotic supplementation [35]	Synbiotic supplements containing Lactobacillus acidophilus, Lactobacillus casei and Bifidobacterium bifidum	↑ (SHBG), plasma NO, ↓ modified Ferriman–Gallwey (mFG) scores and serum hs-CRP			
Dietary soy [36]	0.8 g protein kg^−1^ body weight (35% animal proteins, 35% soy protein and 30% vegetable proteins)	↓ Total testosterone, ↓ triglycerides	↓Triglycerides, body mass index (BMI), fasting plasma glucose, insulin and insulin resistance; ↑ quantitative insulin sensitivity check index	↓ MDA, ↑ NO and GSH	
Probiotic supplementation [47]	Lactobacillus acidophilus, Lactobacillus casei and Bifidobacterium bifidum (2 × 10^9^ CFU/g each) for 12 weeks	↑ Serum SHBG and plasma TAC; ↓ serum total testosterone, mFG scores, serum hs-CRP and plasma MDA		↑ Plasma TAC, ↓ plasma MDA	
Vitamin D-K-calcium co-supplementation [38]	200 IU vitamin D, 90 μg vitamin K plus, 500 mg calcium supplements for 8 weeks	↓ Serum-free testosterone, ↓ luteinizing hormone		↑ Plasma TAC, ↓ plasma MDA	
Vitamin D [48]	4000 IU vitamin D or 1000 IU of vitamin D for 12 weeks	↓ Total testosterone, free androgen index (FAI), hirsutism; ↑ mean change in SHBG		↑ Total antioxidant capacity (TAC)	
Vitamin D [49]	50,000 IU Vitamin D		↓ FPG, insulin, HOMA-IR, estimated B cell function; ↑ quantitative insulin sensitivity check index	↓ Plasma malondialdehyde (MDA) levels	
Vitamin D and [40]	1000 IU vitamin D3 plus 1000 mg Vitamin E		↓ Triglycerides, very low-density lipoprotein (VLDL) cholesterol levels	↑ Serum 25(OH) D and plasma GSH, ↓ MDA	
Vitamin D and omega-3 co-supplementation [41]	50,000 IU vitamin D every 2 weeks plus 2000 mg/d omega-3 fatty acid from fish oil	↓ Serum total testosterone levels; ↑ Beck depression inventory, general health questionnaire scores and depression anxiety and stress scale scores		↓ Serum hs-CRP and MDA, ↑ TAC compared with the placebo	↓ Gene expression of (IL-1) and (VEGF)
Vitamin D and probiotic co-supplementation [42]	50,000 IU vitamin D and 8 × 10^9^ CFU/d probiotic every 2 weeks	↓ Total testosterone		↓ MDA, ↑ TAC and GSH	
MitoQ10 and Vitamin D3 [43]		↓ Estradiol, progesterone, FSH, LH, LH/FSH		↓ MDA and SOD	↓ mRNAs of 3β-HSD, Cyp19a1, Cyp11a1, StAR, Keap1, HO-1 and Nrf2

Abbreviations: FSH: follicle-stimulating hormone, LH: luteinizing hormone, MDA: malondialdehyde, SOD: superoxide dismutase, TAC: total antioxidant capacity, GSH: glutathione, (hs-CRP): high-sensitivity C-reactive protein, (25(OH) D): 25-hydroxyvitamin D, HOMA-IR: homeostasis model of assessment-IR, VLDL: very low-density lipoprotein, SHBG: sex hormone binding globulin, FAI: free androgen index, FPG: fasting plasma glucose, mFG: modified Ferriman–Gallway, TGA: triglycerides, NO: nitric oxide, Hcy: homocysteine. ↓ depressed, ↑ elevated.

## Data Availability

We used PubMed and web of science to screen articles for this narrative review. We did not report any data.

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
