# Peer review of "Effect of Nutritional Supplementation on Oxidative Stress and Hormonal and Lipid Profiles in PCOS-Affected Females"

_nutrients, 2021, doi:10.3390/nu13092938_

Round 1

Reviewer 1 Report

The manuscript presents the description of different factors associated with PCOS and the effects of nutritive supplementation of vitamins, fatty acids and minerals on the markers of oxidative stress caused by PCOS

The conclusion should correspond to the objective, thus please correct the aim of the work (line 47) because not only vitamins but also other nutrients were described in the manuscript.

The good point that review presents many data from numerous studies and these data are organized in the tables. However, the text seems as a catalogue, thus the short conclusions about the main effects of different substances will be useful at the end of each chapter.

The review is difficult to read because of many abbreviations; some of them are not explained, other are detailed several times. It may be useful to make a list of abbreviations at the beginning and use only the short form for the most frequent ones in the text and table, otherwise write the full names for the rare used terms.

A short conclusive figure will be useful

Line 43 – add the reference (“citation”)

L52  IR abbreviation was already explained, remove “insulin resistance”

L88 TAC abbreviation was already explained

L108 Indicate Table 1, remove “list of tables”

In Table 2, it will be useful to add the species (human/animal). If the data on the supplementation reported in the Table 2 was only from human, please indicate this in the table legend. Also, use the shirt abbreviations in the table, and give their descriptions in the legend.

L113 the sentence is not clear. Do you mean that supplementation with vitamin C affects PCOS and the mechanism of this is not known?

L122-132 Please check the use abbreviations through the text, some of them were already mentioned, other are not necessary

L269 Do you mean Effect on gene expression?

In conclusion, do you mean that “ oxidative stress and antioxidant biomarkers can estimate the risk of PCOS severity and cardiovascular events in women”? Remove “the role”

Author Response

All authors thank the reviewers for their intensive and careful review of our manuscript. Here are the changes done according to the recommendations made by the reviewers.

Reviewer 1:

  1. The conclusion should correspond to the objective, thus please correct the aim of the work (line 47) because not only vitamins but also other nutrients were described in the manuscript.

Response: The statement has been changed accordingly throughout the conclusion with the addition of the terms “dietary supplements”.

  1. The good point is that review presents many data from numerous studies and these data are organized in the tables. However, the text seems as a catalogue, thus the short conclusions about the main effects of different substances will be useful at the end of each chapter.

Response: Thank you for the comment. We have listed the major categories with maximum studies into a short paragraph with a chronological description of the study conclusions drawn. Some changes have been made in the text according to the reviewer’s recommendations.

  1. The review is difficult to read because of many abbreviations; some of them are not explained, other are detailed several times. It may be useful to make a list of abbreviations at the beginning and use only the short form for the most frequent ones in the text and table, otherwise, write the full names for the rarely used terms.

Response: We agree with the reviewer about many abbreviations, we have listed every abbreviation used in the narrative as a legend under Table 2.

  1. A short conclusive figure will be useful

Response: A descriptive graphical representation has been provided with the revision.

  1. Line 43 – add the reference (“citation”)

Response: The reference has been added in the previous line.

  1. L52 IR abbreviation was already explained, remove “insulin resistance”

Response: Corrected according to reviewers recommendation

  1. L88 TAC abbreviation was already explained

Response: the full name has been removed

  1. L108 Indicate Table 1, remove “list of tables”

Response: Removed and added the legend

  1. In Table 2, it will be useful to add the species (human/animal). If the data on the supplementation reported in Table 2 was only from human, please indicate this in the table legend. Also, use the shirt abbreviations in the table, and give their descriptions in the legend.

Response: The studies involve only human subjects so it has been updated in the legend for table 2

  1. L113 the sentence is not clear. Do you mean that supplementation with vitamin C affects PCOS and the mechanism of this is not known?

Response: The sentence has been fixed

  1. L122-132 Please check the use of abbreviations through the text, some of them were already mentioned, others are not necessary

Response: This has been fixed.

  1. L269 Do you mean Effect on gene expression?

Response: The gene expression term has been fixed

  1. In conclusion, do you mean that “oxidative stress and antioxidant biomarkers can estimate the risk of PCOS severity and cardiovascular events in women”? Remove “the role”

Response: The statement has been fixed

Reviewer 2 Report

Overall, I thought this review was very interesting and much needed. However, there were several minor changes that need to be made before it can be published.

  • Make sure to define abbreviations the first time you use them. Example:
    • NO is first used in line 76 but not defined till line 125
    • IR is used in line 42 but defined in line 51
  • Also make sure you do not use the same abbreviation for multiple things.
    • GSH is used to define glutathione, glutathione stimulating hormone and reduced glutathione
  • Once you define an abbreviation you do not need to define it again unless its used in a figure or table.
  • Missing citation on line 43 and 84
  • Table 1 is not labeled correctly
  • Add the definition of the abbreviations used in Table 2 at the bottom as a legend
  • Remove bullets from the first study in Table 2 (Folate)
  • For the Melatonin Supplement study, under genes affected do you mean to put tumor necrosis factor in front of alpha?
  • What do you mean by LDL-high-density lipoproteins in line 151?
  • The study description that starts in line 153 thru 157 confuses me. Did they have 3 different diets each with a different animal soy or was the “soy diet” comprise of all three protein sources?
  • Line 238: fix “is which is also can lead”. I feel like either you are missing words or have too many here.

Author Response

All authors thank the reviewers for their intensive and careful review of our manuscript. Here are the changes done according to the recommendations made by the reviewers.

Reviewer 2

  1. Make sure to define abbreviations the first time you use them. Example: NO is first used in line 76 but not defined till line 125, IR is used in line 42 but defined in line 51

Response: The abbreviations have been fixed

  1. Also make sure you do not use the same abbreviation for multiple things.

Response: GSH is used to define glutathione, glutathione stimulating hormone, and reduced glutathione. The two terms—GSH and glutathione—tend to be used interchangeably.

  1. Once you define an abbreviation you do not need to define it again unless it’s used in a figure or table.

Response: This has been fixed throughout the text.

  1. Missing citation on line 43 and 84

Response: This has been fixed.

  1. Table 1 is not labeled correctly

Response: This has been fixed

  1. Add the definition of the abbreviations used in Table 2 at the bottom as a legend

Response: Abbreviations have been fixed and legend has been added

  1. Remove bullets from the first study in Table 2 (Folate)

Response: This has been fixed

  1. For the Melatonin Supplement study, under genes affected do you mean to put tumor necrosis factor in front of alpha?

Response: The TNF alpha-factor has been fixed

  1. What do you mean by LDL-high-density lipoproteins in line 151?

Response: The authors meant LDL, Low-density Lipoprotein, this has been fixed.

  1. The study description that starts in lines 153 thru 157 confuses me. Did they have 3 different diets each with a different animal soy or was the “soy diet” comprise of all three protein sources?

Response: The study comprises a single diet with three different proteins. The confusion has been fixed.

  1. Line 238: fix “is which is also can lead”. I feel like either you are missing words or have too many here.

Response: the grammatical error has been fixed

Round 2

Reviewer 1 Report

Authors have responded to most of the comments.